

# Estimating microplastics emissions from offshore wind turbine blades in the Dutch North Sea

Marco Caboni[1], Anna Elisa Schwarz[2], Henk Slot[3], and Harald van der Mijle Meijer[1]

[1]TNO, Wind Energy Technology, Westerduinweg 3, 1755 LE Petten, The Netherlands
[2]TNO, Climate, Air & Sustainability, Princetonlaan 6, 3584 CB Utrecht, The Netherlands
[3]TNO, Reliable Structures, Molengraaffsingel 8, 2629 JD Delft, The Netherlands

**Correspondence:** Marco Caboni (marco.caboni@tno.nl)

**Abstract.** The continued expansion of offshore wind energy raises concerns regarding the microplastics released from wind turbine blades due to leading edge erosion. Currently, the literature lacks reliable and transparent estimates of microplastic formation and emissions from wind turbines. To bridge this knowledge gap, we employed state-of-the-art models to analytically evaluate the release of microplastics resulting from wind turbine blades' leading edge erosion. This was achieved by integrating

measured offshore weather data with a fatigue-based erosion model. We then applied and extrapolated this methodology to estimate microplastics emissions from all offshore wind turbines installed in the Dutch North Sea and compared these estimates to other sources of microplastics in The Netherlands. Our estimates indicate that microplastic emissions from a modern offshore wind turbine equipped with a polyurethane-based leading edge protection system are approximately 240 grams per year. Using this value, we estimated the current emissions from all wind turbines installed in the Dutch North Sea. Our projections suggest

that the current emissions from Dutch offshore wind turbine blades, amounting to 100 kilograms per year, are approximately one thousand times lower than the total offshore microplastic emissions in the Netherlands when considering other sources, such as the paints and coatings of marine vessels.

## 1 Introduction

To cut greenhouse gas emissions and achieve the Paris Agreement's target of keeping global warming below 1.5°C, the Euro-

pean energy sector must invest in sustainable and renewable energy sources. To reach this target, Europe aims to produce 510 GW of wind energy by 2030. Recent studies have shown that wind turbines can release microplastics (particles smaller than 5 mm) into the environment throughout their life cycle (Hof et al., 2023). A significant cause of material loss in modern offshore wind turbine blades is leading edge erosion (LEE), mainly resulting from the impact of water droplets on the blades. Microplastics in the environment are recognized for their ecotoxicological risks to both wildlife and human health (Gall and Thompson,

2015; Huerta Lwanga et al., 2016; Leslie et al., 2022). As onshore and offshore wind farms are expected to expand to boost Europe's renewable energy share, microplastic emissions from wind turbines could become an additional environmental issue, needing mitigation measures. Quantifying microplastic emissions from wind turbine blades is essential to determine potential mitigation measures. Such quantification has been conducted for sectors like packaging, car tires, and agricultural practices,





which are found to emit substantial amounts of microplastics, up to 0.8 million tons annually on a global scale (Schwarz et al.,
25   2023).

Although extensive research has been conducted on wind turbine LEE (Slot et al., 2015; Mishnaevsky et al., 2021; Verma
et al., 2021) and its impact on wind turbine power production (Bak et al., 2020; Maniaci et al., 2020), a literature survey has
revealed a lack of studies on methodologies to assess microplastic emissions from wind turbine blades. Currently available
studies, reporting values of microplastic formation due to wind turbine blades' LEE, come from government and company
reports or online news items, and are summarized in Table 1. These sources share a common lack of detailed explanations
regarding the methodology, data sources, and assumptions. Without such information, replicating and assessing these studies
becomes challenging, if not impossible. A report by the Dutch National Institute for Public Health and the Environment (RIVM)
(Hof et al., 2023) estimated that each turbine releases between 3 grams and 14 kilograms of microplastics annually. These
estimates, which vary based on blade length and the leading edge protection (LEP) system, were derived from photographs of
eroded blades and communications with unnamed wind turbine manufacturers. A factsheet from NORWEA, the Norwegian
Wind Energy Association, reports an erosion rate of 150 grams per turbine per year (NORWEA, 2021). This information is
reported to have been provided directly by Vestas. On a webpage, Viane (2022) mentions that Vleemo, a Flemish wind farm
developer, reports that wind turbine blades release 640 grams of material per turbine annually. No details are provided about the
methodology used to obtain this figure. Solberg et al. (2021) reports significantly higher microplastic formation compared to
other reviewed sources, estimating a loss of 62 kilograms per turbine annually. This study builds on the work of Pugh and Stack
(2021), who conducted erosion testing on epoxy-glass composite samples using a whirling arm rain erosion test rig, revealing
up to 0.199% mass loss depending on the accumulated rain. Solberg et al. (2021) mistakenly applied this same percentage,
which was derived from a few millimeter samples, to an entire blade (weighing several tonnes). Consequently, this leads to a
disproportionate overestimation of total LEE and the resulting microplastic formation.

**Table 1.** Current estimates of microplastics formation due to leading edge erosion from wind turbine blades.

| Source | mass [kg turbine$^{-1}$ year$^{-1}$] |
| --- | --- |
| Hof et al. (2023) | 0.003-14 |
| NORWEA (2021) | 0.15 |
| Viane (2022) | 0.64 |
| Solberg et al. (2021) | 62 |

The present study aims to tackle the knowledge gap regarding microplastic emissions from wind turbines by employing state-
of-the-art LEE assessment methodologies. The specific goal of this work is to evaluate microplastic emissions resulting from
LEE of offshore wind turbine blades, including those with LEP systems. Using estimates from a representative modern offshore
wind turbine, we aim to calculate the total microplastic emissions from all turbines currently installed in the Dutch North Sea





and project these emissions through 2050. The formation of microplastics during installation, maintenance, deconstruction, or
from other wind turbine components, such as the foundation or cable degradation, is beyond the scope of this study.

Recently measured rain and wind speed offshore data are combined with a fatigue-based erosion model to estimate the
accumulated damage and erosion depth of a representative offshore wind turbine. Detailed information on the weather measurements and erosion model can be found in the methodology section of this paper. As reported in the result section, the
estimated microplastic formation from the reference wind turbine is used to evaluate the contribution of wind turbine blades to
the total microplastic emissions in The Netherlands.

## 2 Methods

### 2.1 Weather measurements

Measured offshore weather data supported this work by providing input to the erosion model. Rain measurements provided
information on the number and size of droplets as well as their fall velocity, while wind speed was correlated with rotor speed,
dictating the impact speed between rain drops and blades. Concurrent measurements of rain and wind at the Dutch North Sea
are scarce. The only available source of measurements are provided by TNO (Caboni et al., 2024) for a relatively short period.
More specifically, over a period of one year, from March 2022 to March 2023, TNO carried out concurrent rainfall and wind
speed measurements at the offshore Lichteiland Goeree (LEG) platform. Figure 1 depicts the exact location of the platform.
Rainfall measurements were carried out by means of an OTT Parsivel$^2$ disdrometer. Wind speed was measured by means of a
Leosphere WindCube V2 LiDAR. Since the current investigation focuses only on rain-induced erosion, measured events with
snow and hail have been filtered out, retaining just rainy intervals. Hailstones, being larger, heavier, and harder than raindrops,
can cause delamination, indentations, and surface cracking depending on their size and falling velocity (Verma et al., 2023).
The impact of hailstones on blade erosion in the North Sea is not yet fully understood, so this topic has been excluded from
the scope of this work. In 2022, the average wind speed at 140 meters above sea level at LEG was 9.3 m s$^{-1}$, which is slightly
lower than the 2015-2022 average of 9.9 m s$^{-1}$ (Vitulli et al., 2023). 2022 was a relatively dry year in the Netherlands, with a
national average rainfall of 729 millimeters (KNMI, 2022). Typically, the country's average annual rainfall is 795 millimeters.

### 2.2 Erosion model

#### 2.2.1 Leading edge erosion modeling

The LEE process is influenced by the fatigue properties of the blade LEP systems and the size and number of droplets impacting the surface at a given speed. Wear particle emissions from the leading edge begin once the incubation period is complete.
Figure 2 illustrates a schematic of erosion depth over time during a droplet erosion test, performed under specific impact conditions such as drop size, impact velocity, and volume concentration. The figure indicates the incubation period and erosion rate.
Methods for accurately calculating the incubation periods and erosion rates of currently applied LEP systems, including the effects of drop size, are limited. The available models are: Slot's physical- and fatigue-based model for the droplet impingement





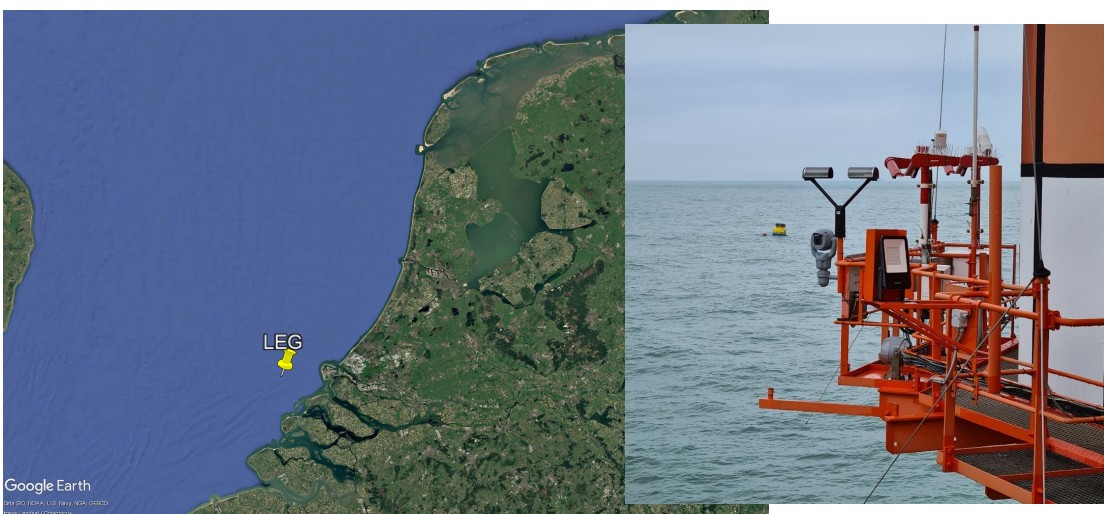

**Figure 1.** A map showing the location of the Lichteiland Goeree (LEG) platform, where concurrent rain and wind speed measurements were conducted (Caboni et al., 2024), and a picture of the OTT Parsivel[2] disdrometer installed on the platform. Map courtesy of © Google Earth.

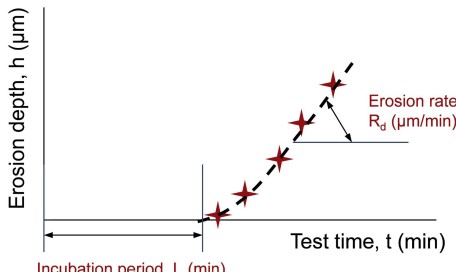

**Figure 2.** The figure illustrates the leading edge erosion process, showing a schematic of erosion depth over time during a droplet erosion test. Figure reproduced from Slot et al. (2015).

80  erosion incubation period (Slot, 2021; Slot et al., 2015), Springer's semi-empirical, fatigue-based model (Springer, 1976), and the "ASTM - Multiple linear regression fit equations" derived from a round robin test program, as detailed in Heymann's work Heymann (1979, 1970). Both Slot's and Springer's models necessitate the material and fatigue properties of the LEP system in use. However, these properties are rarely available or easily determined for most current LEP systems. Consequently, this study estimates the incubation period and erosion rate using the "ASTM - Multiple linear regression fit equations", incorporating a

85  modified drop size dependence (Slot et al., 2025).





### 2.2.2 Incubation period

The "ASTM - Multiple linear regression fit equations" were developed from a comprehensive round robin test program organized by ASTM. Heymann (1979) developed these equations to estimate the incubation period, with each tested material characterized by its normalized incubation resistance number (NOR). Stainless steel AISI 316 was selected as the reference material, assigned a NOR value of 1. Heymann's work suggests that, under reference conditions, the volume of impinged water per unit surface area ($m^3\ m^{-2}$) during the incubation period can be estimated as follows:

$$I_{\mathrm{ref}} = 10^{\log(\mathrm{NOR}) - 5.64 \cdot \log(V_{\mathrm{ref}}) - 2.12 \cdot \log(D_{\mathrm{ref}}) + 15.76} \tag{1}$$

where $V_{\mathrm{ref}}$ represents the reference impact speed (100 m s$^{-1}$) and $D_{\mathrm{ref}}$ denotes the reference drop diameter (1.8 mm). The ratio of the volume of water (per unit surface area) during incubation at the actual drop diameter to that at the reference drop diameter is given by:

$$\frac{I_{V_{\mathrm{ref}},i}}{I_{\mathrm{ref}}} = \left(\frac{D_{\mathrm{ref}}}{D_i}\right)^{1.5} \tag{2}$$

The volume of water per unit surface, impinged over the blade at velocity of $V_k$ and drop diameter $D_i$ during incubation is:

$$I_{ik} = I_{V_{\mathrm{ref}},i} \left(\frac{I_{V_{\mathrm{ref}},i}}{V_k}\right)^{5.64} \tag{3}$$

The number of impacts per unit surface (m$^{-2}$), at velocity of $V_k$, between blade and drops having diameter of $D_i$ during incubation is:

$$N_{ik} = \frac{I_{ik}}{\mathrm{Vol}_i} \tag{4}$$

where $\mathrm{Vol}_i$ is the volume of a rain drop having diameter $D_i$:

$$\mathrm{Vol}_i = \frac{4}{3}\pi \left(\frac{D_i}{2}\right)^3 \tag{5}$$

The measured drop diameters, in the aforementioned campaign at LEG, varied from approximately 0.5 mm to 9 mm. Within this range, none of the currently available erosion models accurately account for the effect of different drop sizes on the incubation period. Even the widely used Springer's model (Springer et al., 1974), along with derived models such as the DNV model reported in DNVGL-RP-0573, does not account for the effect of drop size on the incubation period. Instead, it uses the drop diameter to determine the number of impacting drops on a given area. When evaluating the drop size effect, Slot et al. (2025) observed opposite trends between experiments and Springer's model. Heymann's model considers the effect of drop size, but it is not reliable for relatively small droplets, as these were not included in the aforementioned ASTM test program. To address this, we utilized data from the literature to enhance Heymann's model. Schmitt (1968), Seleznev et al. (2010), Krzyzanowski and Szprengiel (1978), Weigle and Szprengiel (1985) and Bech et al. (2022) reported erosion results from drop impacts on metals, rubber, and elastomers for drop sizes ranging from 0.1 mm to 3.5 mm and impact velocities from 100 m s$^{-1}$ to 224 m s$^{-1}$. Based on these test results, the best representation of the drop size effect was found to be a power of 1.5, as shown in Eq. 2.



### 2.2.3 Maximum erosion rate

Heymann also developed multiple linear regression fit equations for the erosion rate (Heymann, 1979), where each tested material is characterized by its normalized erosion resistance number (NER). Stainless steel AISI 316 was selected as the reference for this resistance number (NER = 1). The derived equation for the maximum erosion rate ($RE_k$) is:

120

$$\log(RE_k \cdot \text{NER}) = 4.78 \log(V_k) - 16.42 \tag{6}$$

$RE_k$ is defined as the erosion depth (m) divided by the volume of impinged water on a certain area (m$^3$ m$^{-2}$). In Heymann's regression equations the drop size is not included and seems to have a minor effect on the erosion rate. NER and NOR values have been determined in this ASTM program for each of the materials tested Heymann (1979). The relation between the NOR and NER values derived for these metals and elastomers is given by:

125

$$\log(\text{NER}) = 0.76 \cdot \log(\text{NOR}) \tag{7}$$

Combining Eq. 6 with Eq. 7 gives the maximum erosion rate as follows:

$$RE_k = 10^{-0.76 \cdot \log(\text{NOR}) + 4.78 \cdot \log(V_k) - 16.42} \tag{8}$$

This equation was employed to estimate the local erosion rate after the incubation period. For cases of non-perpendicular drop impacts, the normal velocity component to the surface should be used. Given the erosion rate, the erosion depth can be

130 calculated as:

$$ED_{ijk} = RE_k \cdot n_{ijk} \cdot \text{Vol}_i \tag{9}$$

where $n_{ijk}$ represents the number of impacts per unit area, during an observation interval, that the a blade section encounters while rotating with a tangential velocity of $TS_k$, involving drops of diameter class $i$ and drop fall velocity class $j$. During the LEG campaign, rain and wind observations were recorded at 10-minute intervals. Assuming the drops are evenly distributed

135 in space, $n_{ijk}$ can be expressed as:

$$n_{ijk} = \frac{n_{ij}}{A \cdot DFV_j} \cdot TS_k \tag{10}$$

Assuming linear damage accumulation (Springer, 1976), the Palmgren-Miner rule was applied to account for the cumulative effects of varying rain and wind speed conditions on the accumulated damage as follows:

$$F = \sum_k \sum_j \sum_i \frac{n_{ijk}}{N_{ik}} \tag{11}$$

140 $F$ represents the cumulative damage, starting at zero at the beginning of the erosion process and reaching one by the end of the incubation period. Following the incubation period, the erosion depth is also calculated cumulatively using the Palmgren-Miner rule as follows:

$$ED = \sum_k \sum_j \sum_i ED_{ijk} \tag{12}$$



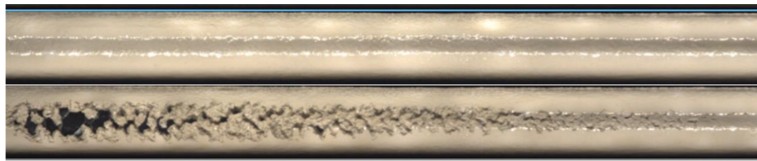

**Figure 3.** Examples of an initial and a tested specimen in the rotating arm tester. This figure is adapted from Hawkins and Nyboe (2019).

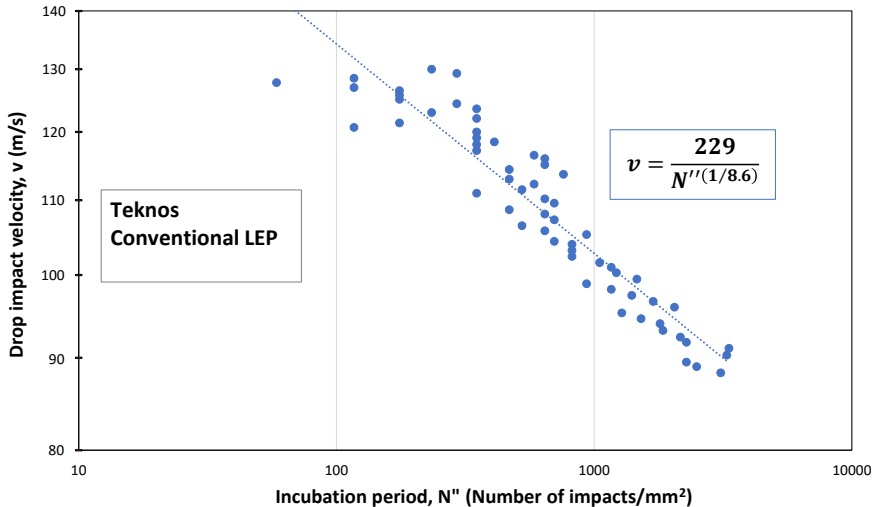

**Figure 4.** The results of periodic inspections of the specimen surface during a rain erosion test on Teknos' LEP system are presented. The fitting equation is displayed.

In this study, we calculated $F$ and $ED$ by summing the contributions from each 10-minute interval of measured rain and wind speed.

### 2.2.4 LEE properties of currently applied LEP systems

A literature study was conducted on recently tested LEP systems in the rotating arm rain erosion tester (RET) setup to determine the NOR value of currently applied systems on wind turbine blades. The RET method, as detailed in DNV-RP-0171 (2008) (DNV, 2021), demonstrates the highest accuracy and reliability when compared to real rain conditions. Figure 3 illustrates an example of a specimen both at the initial stage and after prolonged exposure in the rotating arm test. For example, Figure 4 collects the results of regular inspections of the specimen surface at intervals during a rain erosion test conducted on a Teknos' LEP system. The graph correlates the number of drop impacts at a specific specimen location, thus characterized by a certain tangential velocity, that lead to the end of incubation, defined as the onset of material loss (see Figure 2). The graph illustrates





that as the local velocity decreases, the incubation period, measured by the number of drop impacts, increases. The local drop impact velocity is considered equal to the local specimen tangential velocity. Depending on the type of LEP system sometimes large scatters in obtained data points have been observed. The best fit using a power function for these data points gives the relation between the incubation period and drop impact velocity resulting in the fit constants exponent $m$ (-) and a constant $C$ (m s$^{-1}$) (see Eq. 13).

$$v = \frac{C}{N''^{\frac{1}{m}}} \tag{13}$$

For each LEP system or material documented in the literature, within the present work we established graphs and best-fit functions. The incubation period, expressed as the volume of water per unit area at a velocity of 100 m$^{-1}$, is determined as follows:

$$H = \frac{C\frac{\pi}{6}D^3}{100^m} \tag{14}$$

where $D$ is the drop diameter. The value of $H$ is then compared to the incubation period of stainless steel AISI 316 at the same drop impact velocity to determine the NOR-value of the tested LEP system. The results of this literature inventory are shown in Table 2. For each LEP system, Table 2 depicts the tested mean drop size, $D$, the exponent $m$ and constant $C$ according to Eq. 13. It is noted that the exponent $m$ of these polymeric based LEP systems shows a large scatter, it varies between 5.7 and 18.8 with a mean of 9.5 and a coefficient of variation of 0.45. Table 2 also shows the incubation period, at an impact speed of 100 m s$^{-1}$, of the given LEP system $H$, and the stainless steel AISI 316, $H_{\text{AISI 316}}$. $H_{\text{AISI 316}}$ is determined through Eq. 8 while $H$ through Eq. 14. The last column of Table 2 gives the resulting NOR value of the LEP system. The NOR values of the tested LEP systems vary between 0.001 to 0.033 and most systems have a value below 0.010. For 4 polymer-based LEP systems the NOR value is above 0.010 and for only two systems around 0.030 (note that higher NOR numbers indicate better fatigue properties.).

When interfaces and/or defects are present in the coating system, the incubation period can be shortened by fatigue initiating of subsurface cracks. Also, polymer degradation processes like aging by UV can result in reduction of the incubation period by embrittlement of the exposed LEP system. Especially epoxy has a low resistance to UV degradation and must be shielded from UV radiation by application of a PU top coating (Sánchez, 2023). Currently, more research is needed to understand and quantify the effect of UV on blade degradation. Therefore, this study does not account for UV-related degradation.

## 2.3 Microplastic emission comparison

The microplastic emission estimate from the reference offshore wind turbine that results from our assessment (reported below) was extrapolated to determine the total microplastic formation from all wind turbines installed in the Dutch North Sea. Information on the current and future (until 2050) wind energy assets in The Netherlands, including the number, capacity, specifications, and operation period of wind turbines, was collected from the Global Offshore Wind Farms Database (4C-Offshore,





**Table 2.** Summary of Rotating arm erosion tests (RET) with LEP systems found in literature. The data in this table were calculated by the authors of this study based on underlying results found in the literature.

| Source | LEP system | $D$ [mm] | $m$ [-] | $C$ [m s$^{-1}$] | $H$ [m] | $H_{\text{AISI 316}}$ [m] | NOR [-] |
|---|---|---|---|---|---|---|---|
| Bech et al. (2018) | Coated aluminium | 1.5 | 9.47 | 253 | 11.6 | 12785 | 0.001 |
| DNV (2021) | A13003-H112 (AlMn1Cu) | 2.3 | 7.38 | 349 | 66.4 | 5026 | 0.013 |
| | Coatings 1 & 2 | 2.3 | 16.4 | 183 | 135.3 | 5026 | 0.027 |
| | Coating 3 | 2.3 | 18.8 | 158 | 35.8 | 5026 | 0.007 |
| Herring et al. (2019) | Aluminium (Al3003-H112 ?) | 2.4 | 7.69 | 358 | 126.2 | 4848 | 0.026 |
| Domenech et al. (2020) | Multilayer (S445-178R #2 & #3) | 2.1 | 9.8 | 265 | 70.4 | 6079 | 0.012 |
| | LEP19B-Primer-FillerB-Laminate | 2.1 | 9.52 | 226 | 11.7 | 6079 | 0.002 |
| Herring et al. (2021) | Standard coating (sample 1) | 2.5 | 8.13 | 253 | 15.2 | 4366 | 0.003 |
| | Thicker standard coating (sample 2) | 2.5 | 8.2 | 258 | 18.9 | 4366 | 0.004 |
| | Double standard coating (sample 3) | 2.5 | 13.5 | 183 | 28.6 | 4366 | 0.007 |
| Bech et al. (2022) | PUR/putty/GF-Epoxy | 0.8 | 10.5 | 305 | 28.2 | 54035 | 0.001 |
| | PUR/putty/GF-Epoxy | 1.9 | 10.2 | 245 | 34.1 | 7745 | 0.004 |
| | PUR/putty/GF-Epoxy | 2.4 | 7.78 | 271 | 16.4 | 4805 | 0.003 |
| | PUR/putty/GF-Epoxy | 3.5 | 7.24 | 240 | 12.6 | 2121 | 0.006 |
| Sánchez (2023) | Aeronordic-ID883 | 2.5 | 8.24 | 204 | 2.9 | 4292 | 0.001 |
| | Aeronordic-ID885 | 2.5 | 7.1 | 318 | 30.6 | 4292 | 0.007 |
| | Aeronordic-ID884 | 2.5 | 1.41 | 22661 | 17.5 | 4292 | 0.004 |
| Sánchez (2023) | Aeronordic-ID774, ISO 16474-2: 1000 h | 2.5 | 5.72 | 314 | 5.4 | 4479 | 0.001 |
| | Aeronordic-ID775, ISO 16474-2: 1000 h | 2.5 | 5.72 | 319 | 5.9 | 4479 | 0.001 |
| | Aeronordic-ID783, ISO 16474-3: 7 weeks | 2.5 | 5.74 | 336 | 8.1 | 4479 | 0.002 |
| Hawkins and Nyboe (2019) | Technoblade Repair 9000 | 2.2 | 16.9 | 187 | 197.4 | 5954 | 0.033 |
| | Conventional LEP | 2.2 | 7.69 | 249 | 5.8 | 5954 | 0.001 |
| Tempelis and Mishnaevsky (2023) | PUR/putty/GF-Epoxy | 2.4 | 11.2 | 237 | 112.4 | 4720 | 0.024 |





2024), and through communications with stakeholders and The Netherlands Enterprise Agency (RVO). Using the wind turbine types as provided by the Global Offshore Wind Farms Database, blade length and rotor diameter were obtained from producers websites. For future projects where wind turbine type was not yet known, a linear correlation was tested between capacity of the turbine and blade length (Pearsons correlation test = 0.973). A linear trend line was used to assess the blade length of the unknown wind turbines using Excel v2402. To assess total microplastic formation and emissions from the offshore turbines

in the Dutch North Sea, the emissions from the reference turbine (i.e., IEA 15 MW Reference Wind Turbine (Gaertner et al., 2020)) were linearly scaled with blade length. In other words, if a turbine has blades that are half the length of those on the reference wind turbine, the length of the exposed area that generates microplastics is also halved. The dataset used for this extrapolation is available in supplementaly material.

The resulting total annual microplastic formation from wind turbine blades was compared to all other direct microplastic

emission sources in The Netherlands. Using the existing Material Flow Analysis (MFA) from Schwarz et al. (2023), direct microplastic and macroplastic losses in The Netherlands were mapped. This includes the following sectors: packaging, textiles, automotive, electronic equipment, building and construction, agriculture, primary cosmetic microplastics, and fishery. Micro and macroplastic emissions are modeled to multiple environmental compartments, including ocean environments, roadside and residential soil. Further information on model data and methodology can be further found in Schwarz et al. (2023). In this

assessment, several additions to the model were included compared to the published version. First of all, material flows for paints and varnishes are included as a microplastic source, with paint volume data collected from Kusumgar (2020), applying an average polymer mass of 37% and transfer coefficients from Verschoor (2016). Secondly, microplastic losses from fishing nets were included and estimated at 1% over the lifetime, similar to loss rates to other fiber based products which include textiles and clothing. Furthermore, environmental compartments were updated, aligned to the land cover and land cover change dataset

from the Copernicus institute (Haščič and Mackie, 2018; Copernicus Climate Change Service, 2019). This affects in which type of environment microplastics end up when emitted to air or emitted to water.

## 3 Results

### 3.1 Estimating microplastic release from the reference wind turbine blades

The erosion rate of wind turbine blades due to rain is influenced by rain conditions (such as amount and drop size distribution),

the resistance of the LEP system, and the impact speed between rain drops and blades. As previously mentioned, LEE is driven by the component of the relative velocity between rain drops and the blade surface that is normal to the surface. The impact speed is primarily determined by the blade's tangential velocity. Outboard sections, which have higher tangential velocities than inboard sections, erode faster. Additionally, at a given radial location (characterized by a specific tangential velocity), the aerodynamic curvature of the blade profiles causes different elements along the leading edge to experience varying normal

impact speeds. Consequently, the normal component of the impact speed varies both chord-wise (along the profile shape) and span-wise (at different radii). To determine the erosion rate along the blade, we divided the blade surface into several chord-





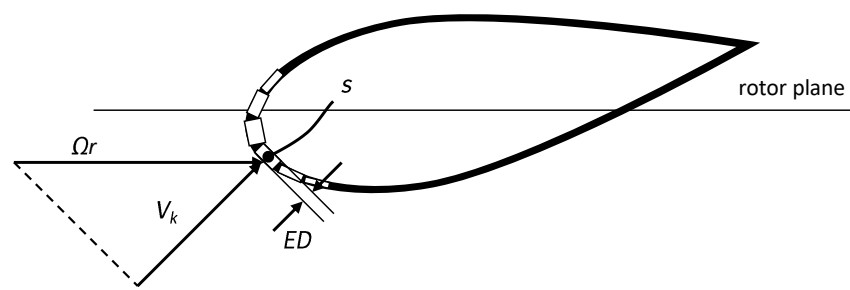

**Figure 5.** A sketch illustrating the parameters used to evaluate the erosion depth of a generic chord-wise element. $\Omega r$ represents the assumed relative velocity between the rain drop and the blade section, where $\Omega$ is the rotor's angular velocity and $r$ is the radial distance of the section from the rotor center. $V_k$ is the component of $\Omega r$ normal to the surface element. $ED$ and $s$ denote the eroded depth and eroded surface area of the generic element, respectively.

wise and span-wise elements. For each of these elements, we calculated the accumulated damage and erosion rate using the normal component of the local impact velocity.

The first challenge is determining the relative velocity between rain drops and the blade. Given the complexity of obtaining
precise estimates of this parameter, certain assumptions were necessary. As mentioned earlier, the relative velocity between rain drops and the blade is primarily influenced by the blade's tangential velocity. However, the terminal velocity of the droplets (which is equal to around 6 m s$^{-1}$ for a 2 mm droplet) and aerodynamic effects also play a role (Barfknecht and von Terzi, 2023). Small droplets are expected to be completely advected by the wind, whereas larger, heavier droplets experience less advection. Using a numerical methodology, Gires et al. (2022) estimated that rain droplets exceeding 0.6 mm in size
experience minimal advection in a turbulent wind field. For simplicity, we assumed that the terminal velocity of the droplets and aerodynamic effects, including advection, are negligible compared to the blade's tangential velocity. In other words, we assumed that rain drops remain stationary in space from an absolute reference frame. Under these assumptions, the relative velocity between rain drops and a blade section is equal to the tangential velocity of that blade section. The tangential velocity, $\Omega r$, is aligned with the rotor plane, as illustrated in Figure 5. Due to the curved shape of the blade profiles, each section around
the leading edge encounters the drops at different normal speeds, $V_k$. We discretized the shape of each blade profile into very small straight elements and calculated the normal component of the tangential velocity for each element, as shown in Figure 5. This component was then used to assess the erosion amount from each element.

In this study, we estimated microplastic formation from the IEA 15 MW Reference Wind Turbine (Gaertner et al., 2020), which features 117-meter-long blades rotating at a maximum tangential velocity of 95 m s$^{-1}$. We assumed that the blades of
the IEA 15 MW Reference Wind Turbine use a polyurethane (PU)-based LEP system with a NOR value of 0.001 and thickness of 0.55 mm. This LEP was selected as it represents the worst-case scenario, having the lowest resistance characteristics among



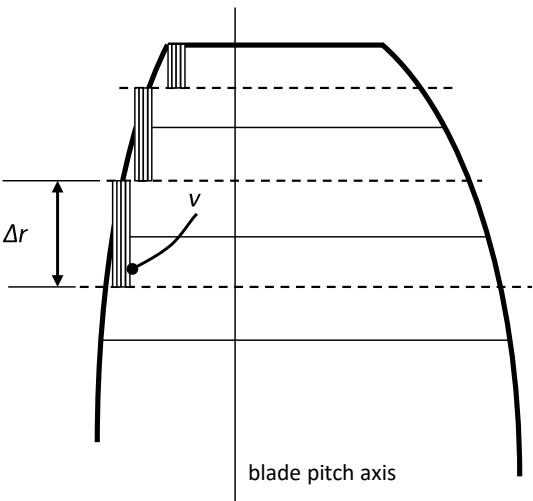

**Figure 6.** A sketch illustrating the parameters used to evaluate the eroded volume of a generic span-wise element. $\Delta r$ represents the span-wise dimension of a generic element, and $v$ denotes the eroded volume from that element.

the LEP systems listed in Table 2. For each blade discretization element, we evaluated the accumulated damage and erosion depth using one year of concurrent rain and wind offshore measurements performed by Caboni et al. (2024). By calculating the reciprocal of the yearly accumulated damage we estimated the incubation period (assuming constant yearly rain and wind

conditions). The eroded surface area shown in Figure 5 is calculated by multiplying the erosion depth by the chord-wise length of each element. The eroded volume depicted in Figure 6 is then calculated by multiplying the eroded surface area by the span-wise dimensions of the elements. The blade span was divided into 100 elements of equal span-wise length, while the profiles were discretized into approximately 200 elements, with finer resolution around the leading edge.

    Figure 7 and Figure 8 illustrate the leading edge discretization elements of the tip airfoil and an airfoil located at 75% of

the blade length from the root, respectively. For each element, the figures show the estimated incubation period and the yearly erosion depth using the aforementioned measured weather conditions. The erosion depth is expressed in millimeters of material removed per year. As expected, the erosion rate is higher around the leading edge, where the normal component of the relative velocity between the droplets and the blade is greater. As we move away from this zone towards the trailing edge, the erosion rate quickly decreases. For both profiles, the highest erosion rate is found on the upper surface. This is due to the section pitch

angle (which is not shown in Figures 7 and 8), rotating the profile counterclockwise and increasing the exposure of the upper section to droplet impact. This is also evident in Figure 5, where the rotor plane, representing the direction of the relative

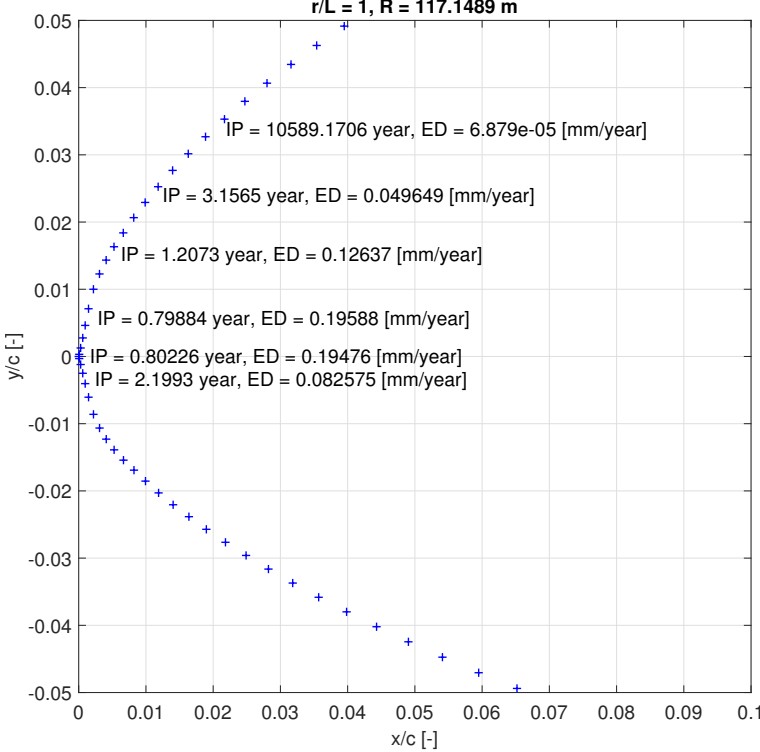

**Figure 7.** The figure illustrates the discretization chord-wise elements of the tip airfoil. For certain elements, the incubation period, $IP$, and the yearly accumulated erosion depth, $ED$, are shown. $r/L$ represents the relative position of the profile along the blade length, while $R$ denotes the radial distance of the airfoil from the blade root. $x/c$ and $y/c$ are the airfoil coordinates, non-dimensionalized using the local chord length.

velocity between the droplets and the blade, intersects the profile on the upper side. This finding aligns with Vimalakanthan et al. (2023), which presents a 3D profile obtained using an optical scanner of a commercial wind turbine (63 m long) blade near the tip. The scan reveals that erosion is predominantly concentrated on the upper surface. Due to confidentiality reasons,
it was not possible to provide operational parameters of this section.

At the most exposed section of the tip, the incubation period is approximately 0.8 years, with an erosion depth of 0.2 mm per year. At the section located 75% of the blade length from the root, the incubation period for the most vulnerable element ends after 5 years, with the erosion depth progressing at a rate of 0.04 mm per year. According to our estimation, erosion reaches a blade section located 75% of the blade length from the root after approximately 5 years. Thus, to prevent erosion from
propagating beyond this section, we assumed that one repair is conducted every 5 years. Under this assumption, we estimated the amount of plastic material eroded by rain from the leading edge of three blades over the assumed 25-year lifetime of the wind turbine. Table 3 summarizes the results. Following blade commissioning and each repair, the accumulated damage resets to zero. No material is removed during the first year, as none of the blade elements reach the incubation period. After two years,



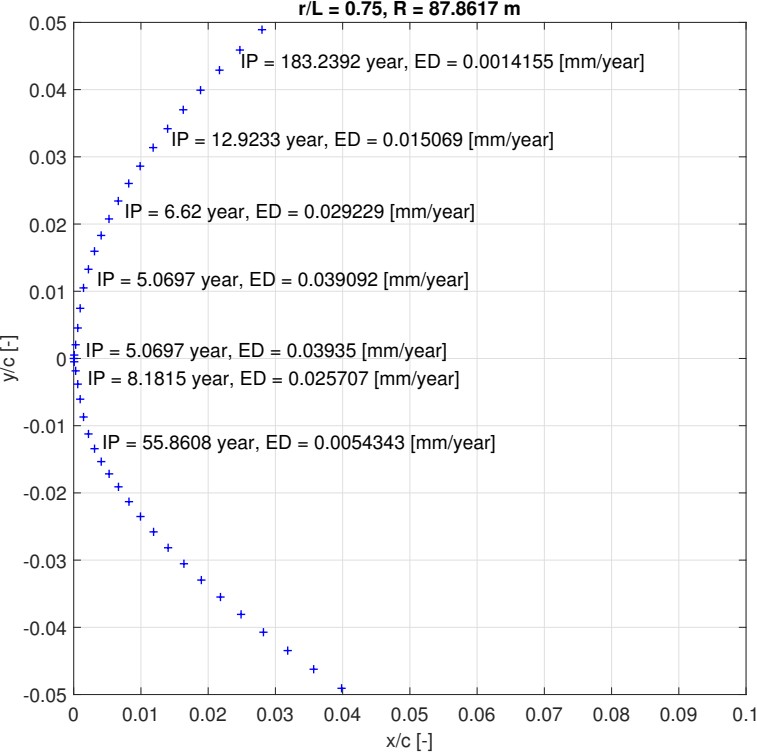

**Figure 8.** The figure illustrates the discretization chord-wise elements of a blade profile located at 75% blade length from the blade root. For certain elements, the incubation period, $IP$, and the yearly accumulated erosion depth, $ED$, are shown. $r/L$ represents the relative position of the profile along the blade length, while $R$ denotes the radial distance of the airfoil from the blade root. $x/c$ and $y/c$ are the airfoil coordinates, non-dimensionalized using the local chord length.

material begins to be released from the tip. The erosion then progresses inboard, increasing the amount of material lost. Once
an erosion depth of 0.55 mm (the assumed thickness of the LEP system) is reached at a specific element of the profile leading edge, we assume that no further material can be released from that area. Indeed, by ensuring that operations and maintenance are effectively carried out, the appropriate maintenance and inspection strategy prevents excessive damage beyond the LEP system. Over the lifetime of the turbine, it is estimated that the total volume of material lost from three blades is about 0.006 $m^3$. With an average PU density of 1 g $cm^{-3}$, this equates to a mass loss of approximately 6 kg. This results in an average
production of 240 grams of microplastics per turbine each year.

## 3.2 Blade microplastic emission and impact analysis results

In the previous section we estimated that the IEA 15 MW Reference Wind Turbine (Gaertner et al., 2020), which has 117-meter-long blades, releases an average of 240 grams of microplastics annually based on measured offshore weather conditions. Using this figure, we estimated the microplastics release from the different wind turbines installed in the Dutch North Sea by scaling





**Table 3.** The table estimates the volume and mass of microplastics released due to leading edge erosion from the blades of the IEA 15 MW Reference Wind Turbine, which features a polyurethane LEP system. It shows microplastic release over time and throughout the turbine's lifetime, assuming one repair every five years.

| year | volume [$m^3$ turbine$^{-1}$ year$^{-1}$] | mass [kg turbine$^{-1}$ year$^{-1}$] |
|------|------|------|
| 1 | 0 | 0 |
| 2 | 0.000123068 | 0.123068048 |
| 3 | 0.000260566 | 0.260565884 |
| 4 | 0.000372744 | 0.372743561 |
| 5 | 0.000411239 | 0.411239404 |
| 6 | 0 | 0 |
| 7 | 0.000123068 | 0.123068048 |
| 8 | 0.000260566 | 0.260565884 |
| 9 | 0.000372744 | 0.372743561 |
| 10 | 0.000411239 | 0.411239404 |
| 11 | 0 | 0 |
| 12 | 0.000123068 | 0.123068048 |
| 13 | 0.000260566 | 0.260565884 |
| 14 | 0.000372744 | 0.372743561 |
| 15 | 0.000411239 | 0.411239404 |
| 16 | 0 | 0 |
| 17 | 0.000123068 | 0.123068048 |
| 18 | 0.000260566 | 0.260565884 |
| 19 | 0.000372744 | 0.372743561 |
| 20 | 0.000411239 | 0.411239404 |
| 21 | 0 | 0 |
| 22 | 0.000123068 | 0.123068048 |
| 23 | 0.000260566 | 0.260565884 |
| 24 | 0.000372744 | 0.372743561 |
| 25 | 0.000411239 | 0.411239404 |
| $\sum$ | **0.005838084** | **5.838084485** |

the reference turbine's estimate linearly according to blade length. Figure 9 depicts the estimated release of microplastics into the oceanic environment from offshore wind turbines located over the Dutch North Sea between 2024 and 2050. Between 2024 and 2050, microplastic emissions are expected to rise due to the increasing installed capacity of wind turbines in the North Sea, growing from the current 100 kilograms per year to a peak of around 650 kilograms per year by 2041. Approximately, there is

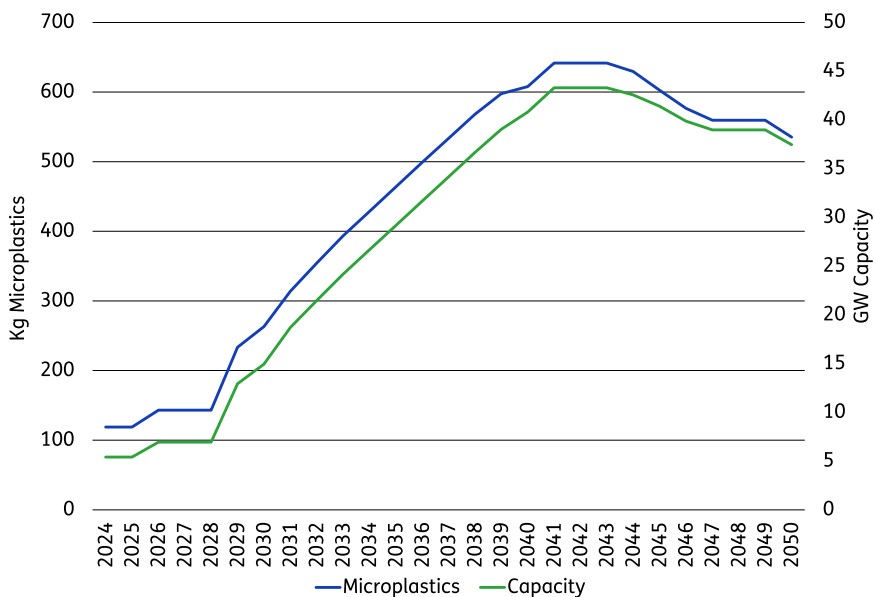

**Figure 9.** Annual microplastic emissions and the associated yearly capacity from Dutch offshore wind turbines in the North Sea.

a microplastic release of $0.02 \, \mathrm{kg} \, \mathrm{MW}^{-1}$ installed capacity, with a slight decrease in the future, from 0.021 kg to 0.014 kg. This

is a direct result of improved capacity of turbines, allowing longer wind turbine blades which requires fewer wind turbines in total. This results in relatively lower material input for the turbines. The decrease in capacity after 2041, as shown in Figure 9, is due to the decommissioning of wind farms that have reached the end of their operational life. Potential repowering of existing wind farms or the development of new wind farm areas after 2030 are not included in the data and are therefore not represented here.

When comparing the yearly offshore wind turbine emissions to other microplastic emission sources in The Netherlands, wind turbines only contribute to small amounts to the total emissions. In 2017, around 23 kilotonnes of microplastics were directly released into the environment in The Netherlands, with approximately 70% of that total originating from car tires (see Figure 10). This value excludes any macroplastic emissions (> 5 mm), which includes littering and improper waste management practices, such as dumping of (plastic) waste. Figure 10 indicates that microplastic losses from Dutch fisheries are assessed

to be minimal. Additionally, we utilized the MFA to evaluate and compare microplastic emissions with all direct emissions to oceanic environments from The Netherlands. This assessment includes emissions from macroplastics but excludes any transport from other compartments, such as river inflows. In 2017, a total of 155 tonnes of micro and macroplastics were emitted into the Dutch North Sea. These direct plastic emissions originated from both the paints and coatings of marine vessels



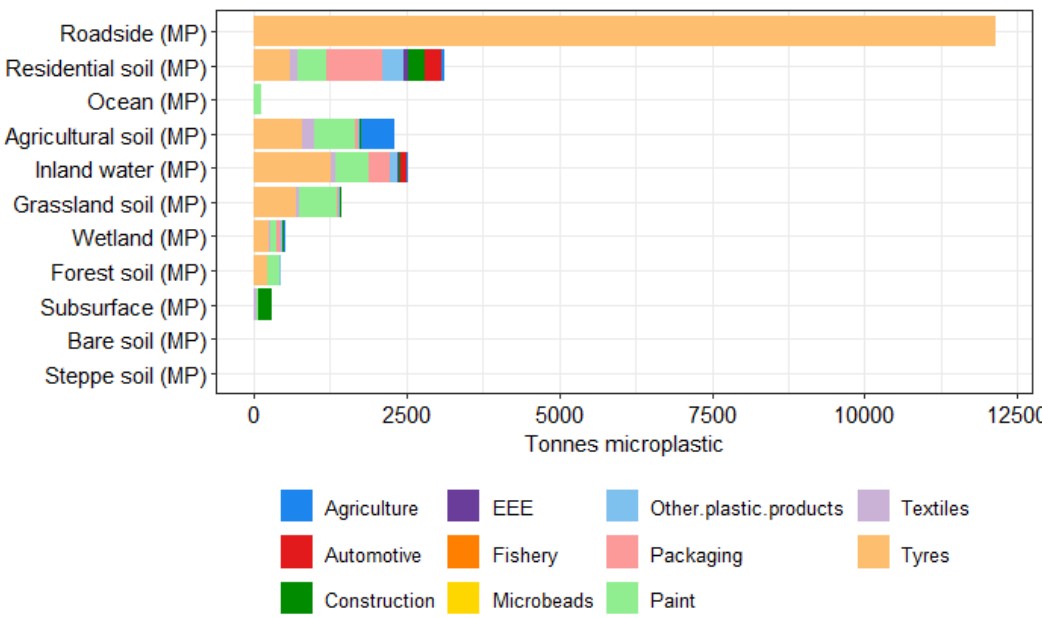

**Figure 10.** Microplastic emissions to the environment for The Netherlands in 2017, assessed through material flow analysis. Model source: Schwarz et al. (2023).

and the direct loss of fishery equipment. Therefore, the estimated 0.1 tonnes currently emitted from offshore wind turbine
blades in the Dutch North Sea represent less than 1‰ (one per mille) of the total annual plastic offshore emissions in The
Netherlands.

## 4   Discussions

Our estimate of 240 grams per turbine per year is on the lower end of the existing range of available estimations (see Table 1),
which spans from 3 grams to 14 kilograms per turbine per year (excluding the erroneous estimate by Solberg et al. (2021)). At
240 grams, our estimate is rather close to the 150-gram estimate provided by Vestas (NORWEA, 2021). The aforementioned
lack of information on the estimate in the literature prevents us from further investigating the reasons for the differences.

As observed and compared in the results section, it was found that the contribution of microplastics emission from offshore
wind turbine blades in the Dutch North Sea is minimal. This is also the case when only the direct emissions sources into oceanic
environments are considered, which include fishery and paint as sources. Furthermore, it can be assumed that the microplastics
emitted from the turbine blades will sink quickly to the ocean floor. As the polymer coating on the wind turbines is PU, which
has a density higher than seawater. This additionally means that the exposure time of the microplastics in the higher water
column is minimal, and with that its potential impacts on species living here. However, it is likely that degradation rates of
these microplastics at the seafloor are slow, due to absence of UV in these areas. Local accumulation of PU microplastics can





occur, which can negatively affect species living on the ocean floor. Comparing the long-term impacts of microplastic release
with other effects from offshore wind installations, such as the impact on migratory birds and disturbances to the ocean floor
during installation, maintenance, and decommissioning, remains a significant challenge. Future studies should aim to compare
and address the biodiversity impacts of these activities.

The calculations executed in this study are related to offshore wind turbines and are also only applied on offshore installed
capacity. However, wind turbines are also installed onshore, where microplastics are emitted closer to human environments. In
The Netherlands, onshore capacity is lower compared to offshore capacity. Additionally, often tip speed on onshore installation
is lower compared to offshore systems to reduce noise pollution on land. However, this also results in lower investments in LEP
technology on onshore systems. Still, it can be safely assumed that the contribution of microplastic emissions from onshore
systems is low, especially compared to other emission sources of microplastics to terrestrial environments, such as from car
tyres, which is significantly higher. This underlines the low contribution of wind turbines to total microplastic emission to the
environment. However, other problems can occur due to degradation without proper LEP, such as the potential emissions of
Bisphenol A by absence, or without maintenance, of LEP (NORWEA, 2021).

It is important to note that this study does not include microplastic emissions from other wind turbine components and
sources, such as foundation coatings and electrical grid cables. Additionally, plastic wear and tear during production, instal-
lation, operation, maintenance, and decommissioning are excluded. The methodology for assessing LEP degradation assumes
proper application and degradation under modeled LEE conditions. However, faults and issues during LEP application and
maintenance, such as repairs, can accelerate LEP erosion.

Our estimations, made using state-of-the-art LEE methodology, assume that all offshore turbines use the same materials,
operate similarly, and experience the same environmental conditions. The short-term weather conditions used in the assessment
were recorded during a relatively dry period with wind speeds slightly below average, leading to an underestimation of the
degradation rate. Despite this, the calculation offers a first scientific approach to assess the scale of microplastic emissions
from wind turbines in oceanic environments.

## 5 Conclusions

In this study, we assessed the total quantity of microplastics emitted by wind turbines currently operating in the Dutch North
Sea and projected this data through 2050. Our estimates indicate that the release of microplastics into the offshore environment
is currently around 100 kg per year, with projections suggesting it could rise to approximately 650 kg by 2041. Therefore the
estimate of microplastics currently emitted from offshore wind turbines in The Netherlands account for a very small portion of
the total microplastics released offshore in The Netherlands, specifically less than 1‰ (one per mille).

This study indicates that the microplastics released from wind turbines due to rain-induced leading edge erosion are neg-
ligible. Although our estimates are based on state-of-the-art methodologies, the calculation chain is subject to significant
uncertainty. In our work we used short-term offshore measurement to estimate the impact of erosion. Future research should
focus on understanding long-term rain and wind conditions, considering the increase in extreme events due to climate change.





Furthermore, future research should enhance the representativeness of rain erosion tests and evaluate the impact of UV exposure and hailstones on blade erosion. In our study, we estimated microplastic formation solely from blade rain-induced erosion at the leading edge. However, microplastics can also be released into the environment during maintenance activities, such as
blade repairs. Future research should aim to quantify this source.

*Author contributions.* **M. C.**: Drafted the paper, developed the methodology for evaluating leading edge erosion using weather data, and conducted the analysis of the results. **A. S.**: Performed the material flow analysis, and conducted the analysis of the results. **H. S.**: Developed the erosion model. **H. van der M. M.**: Collected and integrated information from the Global Offshore Wind Farms Database, stakeholders and The Netherlands Enterprise Agency (RVO).

*Competing interests.* The authors declare that there are no conflicting interests.

*Acknowledgements.* This research was supported by TNO's knowledge innovation project program (KIP) 2023 and, within the PROWESS project, The Netherlands Enterprise Agency (RVO), part of the Dutch Ministry of Economic Affairs, under grant number HER+-00900701.



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
