# Peer review of "Estimating microplastics emissions from offshore wind turbine blades in the Dutch North Sea"

_Wind Energy Science, 2024_

## Author Comment (AC2)

**Response to Referee 1**

**Estimating microplastics emissions from offshore wind turbine blades in the Dutch North Sea (wes-2024-175)**

Dear Reviewer,

Thank you for reviewing our article. We have carefully addressed your comments, and the details are provided below.

Sincerely,

Marco Caboni, Anna Elisa Schwarz, Henk Slot, and Harald van der Mijle Meijer

First, some very minor details:

- I would add in the abstract that the estimate of 240 g per turbine is for a 15 MW turbine.

We have included this information in the abstract.

- In Table 1, I would prefer it if it was indicated that the Solberg et al., (2021) estimate is based on errors. It is mentioned in the main text, but it is worth repeating in case readers copy the table.

Done. The caption of Table 1 now contains this information.

- In the SI excel file, the word "length" is misspelled.

Corrected.

Some larger points to further improve the manuscript:

A comment on the structure of the methods and results section:

Throughout the methods and results section, calculation methods and their results are interwoven. I was a bit lost when reading the methods section, as some important information on the calculation methods are not discussed until the results section. It would be better to either present methods and assumption separately, or to change the main headings to better represent that the text in chapter 3 includes both methods and results.

We agree on this. We have moved the methodology for assessing erosion levels on the leading edge of wind turbines from the Results section to the Methods section. We have created and new paragraph entitled "Assessment of erosion levels on the leading edge of wind turbines".

I think the discussion could be more in depth, some points:

**Line 298 - 301** *"Our estimate of 240 grams per turbine per year is on the lower end of the existing range of available estimations (see Table 1),*

*which spans from 3 grams to 14 kilograms per turbine per year (excluding the erroneous estimate by Solberg et al. (2021)).. At*

*240 grams, our estimate is rather close to the 150-gram estimate provided by Vestas (NORWEA, 2021). The aforementioned*

*lack of information on the estimate in the literature prevents us from further investigating the reasons for the differences"*

It is worth mentioning that the upper estimate of 14 kg is also a deliberate overestimation by the original authors, which assumed the entire length of the blade eroded (instead of the more realistic 25% in this study). Furthermore, some of the estimates are for blades without leading edge protection.

Thank you for bringing this to our attention. We have incorporated this information into the literature review section of the introduction.

I agree that comparison to results of other sources is otherwise difficult, but the authors could spend some time discussing the impact of their own assumptions on the results.  E.g. in the current study it was assumed all wind turbines have LEP. The authors could discuss how valid of an assumption this is for wind turbines in the Dutch North Sea, especially the older existing parks.

This comment is addressed by including the following paragraph in the Discussion section:

*"Our study assumed that all wind turbines have LEP systems. However, the actual situation is more complex. In the Netherlands, no special LEP systems were used on the first generation of offshore turbines installed in 2006-2007 (33 Vestas V90-3MW wind turbines at Princess Amalia Wind Farm and 60 Vestas V80-2MW wind turbines at Offshore Windpark Egmond aan Zee). The blade leading edges of these turbines are equipped with a standard protective blade coating system (gel coating, UV resistant). On modern offshore wind turbines placed in the North Sea, an LEP coating and/or soft shell system is used instead. Therefore, the resistance characteristics of the LEP system assumed in this study overestimate those of the first-generation wind turbines and better match those of the new turbines. The authors lack the necessary information to estimate the NOR values of the actual LEP systems for both the old and new wind turbines."*

I also think it would be valuable to calculate a range of emission values, for different LEPs, instead of a single worst-case value. I believe the required input (table 2) is there already.

We agree that this discussion is valuable for the paper. In the Discussion section, we outlined the impact of the LEP systems' NOR values on the emissions as follows:

*"In our study, we estimated the emissions of microplastics from a turbine with blades featuring a PU-based LEP system. We assumed a NOR value of 0.001 for this LEP, representing the worst-case scenario due to its lowest resistance characteristics among the LEP systems listed in Table 2. The NOR value has a significant impact on both the incubation period and the erosion rate. According to Eq. 1 and Eq. 2, the NOR value is inversely proportional to the accumulated damage and, consequently, directly proportional to the incubation period. Doubling the NOR value results in doubling the incubation period. Based on the current assumptions (NOR = 0.001), the most exposed section of the blade, specifically the leading edge of the tip section (Figure 7), has an incubation period of approximately 0.8 years. Therefore, using an LEP system with a NOR value greater than around 0.03 (while maintaining the assumptions on rain and wind speed) would extend the incubation period of the most exposed section to 25 years, matching the turbine's lifetime and effectively eliminating microplastics release. Table 2 lists one LEP system that theoretically has such high resistance that, according to the calculation methodology used in this study, it would solve the issue of microplastics release from blades. However, such theoretical results need to be validated in the field, which is beyond the scope of this paper. The NOR values in Table 2 were determined in laboratory environments, allowing for perfect application of the LEP and not accounting for aging effects like UV exposure. However, in the field, LEP systems operate in challenging marine environments and are exposed to aging effects, which likely impact their resistance characteristics."*

**Line 304- 308** *"Furthermore, it can be assumed that the microplastics emitted from the turbine blades will sink quickly to the ocean floor. As the polymer coating on the wind turbines is PU, which has a density higher than seawater. This additionally means that the exposure time of the microplastics in the higher water column is minimal, and with that its potential impacts on species living here. However, it is likely that degradation rates of these microplastics at the seafloor are slow, due to absence of UV in these areas. Local accumulation of PU microplastics can occur, which can negatively affect species living on the ocean floor."*

I don't think the statements on the environmental fate of the microplastics are well supported by the current text. The authors claim the polymer will sink quickly due to the higher density of PU than that of seawater. However, the density of PU is said to be 1 g/cm3 in line 269, which is the same density as that of water. The section would benefit from some references on environmental fate of microplastics in seawater.

Thank you for pointing this out. We acknowledge that the two parts are indeed contradictory. Firstly, we have provided a range of densities for PU-based coatings in the Results section as follows:

*"The density of PU-based coatings for wind turbines varies depending on the specific material composition. Our best estimates range between 0.9 g cm^-3 and 1.1 g cm^-3. Given this range, we used a representative value of 1 g cm^-3 to estimate the mass of material lost from the volume lost."*

Secondly, we have discussed the impact of the coating's density on the fate of microplastics in the Discussion section as follows:

*"The fate of microplastics released from wind turbine blade LEP systems at sea is influenced by their density, which varies depending on the specific material composition. PU-based coatings have densities ranging from 0.9 g cm^-3 and 1.1 g cm^-3. Microplastics generated from PU-based coatings with a density greater than that of seawater (which ranges from 1.02 g cm^-3 and 1.03 g cm^-3) will quickly sink to the ocean floor. Consequently, the exposure time of these microplastics in the upper water column is minimal, reducing their potential impact on species living there. However, microplastics generated from PU-based coatings with a density lower than that of seawater will float on the sea surface, posing a greater potential impact on living species. The authors lack specific data regarding the density of LEP systems currently in use on wind turbine blades installed in the Dutch North Sea. Although PU is rarely found in ocean surface waters (Lebreton et al., 2018), the authors believe that further research is needed to assess the fate of microplastics generated from the LEP systems used in actual wind turbine blades, through measurements taken around wind turbine installations."*

Lastly, the introduction mentions the necessity of this work for the determination of potential mitigation measures. The discussion would benefit from a reflection on this point. What can be learned from the current study in regards to minimizing microplastic emissions from wind turbines? Or what kind of further research would be required to further investigate this?

Firstly, in the Introduction section, we have outlined the following mitigation strategies:

*"Quantifying microplastic emissions from wind turbine blades is essential to determine potential mitigation measures, such as the development of anti-erosion coatings (Mishnaevsky et al., 2023), the implementation of effective maintenance practices for erosion prevention, and the application of erosion-safe modes (Bech et al., 2018), which involve reducing the turbine's rotational speed during periods of heavy rain and high wind speeds."*

Secondly, in the Conclusion section we addressed future research as follows:

*"Our research indicates that the release of microplastics from blades due to rain-induced LEE is negligible compared to other sources. However, the author emphasizes the need for future research to develop mitigation strategies, such as new erosion-resistant coatings, maintenance practices for erosion prevention and turbine operational erosion-safe modes that could potentially eliminate these emissions entirely. Future research is also necessary to address the unresolved aspects of this study. As previously mentioned, our analytical estimate relies on numerous assumptions and models that need validation. In our work we used short-term offshore measurement to estimate the impact of erosion. Future research should focus on understanding long-term rain and wind conditions, considering the increase in extreme events due to climate change. Furthermore, future research should validate and enhance the representativeness of rain erosion tests and evaluate the impact of UV exposure and hailstones on blade erosion. Additionally, our research did not consider other sources of microplastics from various wind turbine components or maintenance activities such as repairs. These sources, which we currently cannot quantify, could be significant. Furthermore, we highlight the need for additional research to assess the fate of microplastics generated from the LEP systems used in modern wind turbines, through measurements taken around wind turbine installations. Additionally, our research did not evaluate the toxicity of wind turbine polymers released as microplastics into the environment, nor is it clear how these polymers decompose in a marine environment."*